# Diffusion Tensor Imaging in Syringomyelia Secondary to Chiari Malformation in Cavalier King Charles Spaniel—A Preliminary Study

**DOI:** 10.3390/ani12233405

**Published:** 2022-12-02

**Authors:** Marcin Adam Wrzosek, Aleksandra Ewa Banasik, Karolina Owsińska-Schmidt, Anna Zimny

**Affiliations:** 1Department of Internal Diseases with a Clinic for Horses, Dogs and Cats, Faculty of Veterinary Medicine, Wrocław University of Environmental and Life Sciences, 50-366 Wrocław, Poland; 2Department of General and Interventional Radiology and Neuroradiology, Wroclaw Medical University, 50-551 Wrocław, Poland

**Keywords:** diffusion tensor imaging (DTI), Chiari malformation (CM), syringomyelia (SM), Cavalier King Charles Spaniel (CKCS), magnetic resonance imaging (MRI)

## Abstract

**Simple Summary:**

In our research, we discussed the problem of Chiari-like malformation—syringomyelia syndrome (CM-SM syndrome). It is challenging to determine the harmfulness of the disease as the severity of clinical symptoms is often not commensurate with MRI results. For the same reason, another challenge is a preventive prediction of the disease. In this study, we used the modern technique magnetic resonance diffusion tensor imaging (MR-DTI) to assess the microstructural degree of spinal cord damage in the course of syringomyelia. We found a difference in two DTI parameters: fractional anisotropy (FA) and apparent diffusion coefficient (ADC) between non-symptomatic and symptomatic Cavalier King Charles Spaniel dogs. The use of DTI imaging in the MRI evaluation of CM-SM patients carries potential value in the development of a clinically useful protocol for an objective assessment of the spinal cord and to understand what processes lie at the basis of many diseases, the diagnosis of which is currently difficult. Tests performed on animals may contribute to progress in the diagnosis of CM-SM in humans.

**Abstract:**

Syringomyelia secondary to Chiari-like malformation (so-called CM-SM syndrome) is a common disorder in Cavalier King Charles Spaniels (CKCS) that is diagnosed using standard structural MRI, though imaging findings often do not correlate with the severity of clinical symptoms. Diffusion tensor imaging (DTI) is a technique that defines subtle microstructural changes in the course of many brain and spinal cord diseases, that are not visible on standard MRI. The aim of the study was to identify the correlation between the presence of clinical symptoms and DTI parameters, such as apparent diffusion coefficient (ADC) and fractional anisotropy (FA) within the spinal cord in the course of CM-SM. Study subjects included 18 dogs, CKCS with MRI-confirmed SM (SM group), and 12 CKCS dogs without SM (non-SM group). The SM group was divided into SM-symptomatic group (*n* = 8) and SM-asymptomatic group, *n* = 10). All dogs underwent same clinical and neurological assessment followed by MRI examination. All MRI studies were performed on a 1.5T MRI scanner. The MRI spine protocol included: transverse and sagittal T2-weighted images followed by DTI performed in the sagittal plane. The measurements of FA and ADC values were performed manually using the region of interest (ROI) method at the level of three intervertebral discs between C1 and C4. Notable differences in age and body weight were found. No significant differences in FA and ADC values between the SM and non-SM groups were found, but between non-SM, SM-asymptomatic and SM-symptomatic groups significant differences were found in ADC values in all three ROIs and in FA values in ROI-1 and ROI-3. SM-symptomatic dogs compared to non-SM, showed decreased FA value in ROI-1 and ROI-3 also increased ADC value in ROI-1, ROI-2 and ROI-3. SM-symptomatic dogs compared to SM-asymptomatic showed also decreased FA value in ROI-1 and ROI-3, and also increased ADC value in ROI-1, ROI-2 and ROI-3. The results suggest that the values of DTI parameters correlate with the severity of clinical symptoms in the course of CM-SM in animals. The use of DTI evaluation of CM-SM patients carries a potential value as a clinically relevant protocol for an objective assessment of the spinal cord.

## 1. Introduction

Syringomyelia (SM) is defined as an abnormal fluid cavity within the spinal cord. SM secondary to Chiari-like malformation (so-called CM-SM syndrome) is a severe spinal deformation leading to neurological symptoms. In dogs, this neurological disorder is most commonly observed in Cavalier King Charles Spaniel (CKCS) [1]. Several studies have shown that with age, the percentage of dogs that have visible MRI changes of CM-SM syndrome increase; however, the animals are often asymptomatic [2]. The most commonly reported clinical symptoms include vocalization, spinal pain, touch aversion, altered emotional state, sleep disturbance, head-scratching and phantom scratching [1,2]. CM-SM syndrome also occurs in humans. The incidence of CM-SM in humans is estimated at 8.4 new cases/year/100,000 people [3]. The imaging method of choice in the diagnosis of CM-SM is MRI. The most common MRI findings include occipital bone hypoplasia, central canal dilatation and syringomyelia. Owing to the non-specific nature of the disease, it is difficult to determine its severity based on clinical symptoms, especially when they have subtle characteristics in both human and veterinary medicine [1,2,3,4,5,6,7,8,9,10,11,12,13,14]. The reason for the low correlation between the severity of clinical symptoms and structural changes in the spinal cord visible on MRI is still unknown. Another challenge is a preventive prediction of disease in asymptomatic breeding dogs. Therefore, further research using advanced MRI sequences such as diffusion tensor imaging (DTI) is justified.

Diffusion tensor imaging is an MRI modality which can provide a detailed assessment of the intrinsic spinal tracts and a better understanding of how tissue damage causes clinical deficits. It is a non-invasive MRI technique which is more sensitive to microstructural changes than conventional MR images and is able to show abnormalities within the spinal cord which are not visible on standard structural MR images. It measures microstructural characteristics of water diffusion within the nervous tissue [15,16,17,18,19,20,21,22]. Diffusion tensor imaging is a method that has been increasingly used in the assessment of myelopathies, such as syringomyelia, and degenerative myelopathy or traumatic spinal injuries [19]. This technique is ideal for assessing disorders in which there is a loss of white matter integrity, demyelination, and the associated changes in the movement of water molecules. Nervous tissue, owing to its ordered structure, is ideally suited for examination using the sensitive DTI technique. Changes in fractional anisotropy (FA) and apparent diffusion coefficient (ADC) values derived from DTI enable the viewing of microstructural damage that is not detectable using standard structural MR images [18]. The ADC and FA parameters provide information on water diffusion in the extracellular space. FA values correlate with the white matter integrity, reflecting coherence, organization, and density of the fibre bundles in the spinal cord [2,15], whereas ADC values indicate the amount of water diffusion in the tissue volume.

DTI parameters have been used for detecting subtle damage to the spinal cord in the course of syringomyelia in humans, but those reports are limited and based on a small number of patients and a small size of SM [6]. Published data suggest that as the integrity of the white matter fibres in the spinal cord deteriorates, ADC values increase, and FA values decrease [6,7,8,9,10,11,19,20]. Based on the above, we hypothesized that in the course of SM, the ADC value will increase, and the FA value will decrease. Based upon the latter reports, we hypothesized that ADC and FA values may correlate with clinical symptom development in a course of CM-SM in CKCS and be of prognostic use.

To our knowledge, this is the first report evaluating DTI parameters in dogs with syringomyelia use in the clinical setting.

## 2. Materials and Methods

### 2.1. Subjects

This is a prospective, clinical study, subjects consisted of CKCS which were veterinary patients in the Department of Internal Disease with Clinic for Horses, Dogs and Cats Faculty of Veterinary Medicine at Wroclaw University of Environmental and Life Sciences in Poland from April 2020 to June 2022. The research did not require the approval of the Local Ethics Committee. The owners signed their permission for their pets to undergo an MRI examination, the research was performed on the images and data obtained from clinical trials.

All animals were client-owned dogs and all owners agreed that their pet’s imaging results could be used in this research. Dogs with ear diseases, dermatological, orthopaedic problems were excluded from the study, because these disorders may present similar clinical symptoms to CM-SM syndrome. Study material included 18 dogs with MRI-confirmed SM (SM group) (age 6–70 months, weight 3.5–9.1 kg, 6 males, 12 females) and 12 healthy dogs without SM (non-SM group—control group) (7–51 months, weight 4–9.5 kg, 5 male, 7 female). In the SM group, 8 dogs were found to have clinical symptoms (SM-symptomatic) and 10 dogs had no clinical symptoms (SM-asymptomatic). All animals underwent the same predefined study protocol that included a detailed clinical and neurological examination followed by predefined MRI examination protocol. The neurological examination consisted of mental status and behaviour assessment, attitude/posture, gait, abnormal movements, postural reactions (proprioceptive positioning, hopping, hemiwalking), cranial nerves assessment, spinal reflexes (withdrawal reflex, patellar reflex, cranial tibial reflex, extensor carpi radialis reflex, perineal reflex, cutaneous trunci reflex), palpation of a head, neck and spinal cord.

Based on interviews with the owners and neurological examination, the dogs were assigned to the SM-symptomatic or SM-asymptomatic groups. The following symptoms were reported: abnormal behaviour, “phantom scratching”, fly- catching, ear flapping, neuropathic pain, and neurological deficits such as thoracic limb weakness and muscle atrophy, as well as pelvic limb ataxia and weakness [1,2,15,22]. Ear diseases, dermatological, orthopaedic problems, infection or other non-neurological problems were ruled out in all dogs enrolled in this study. The neurological examination was conducted by authors (AB—DVM with 2 years’ experience, KOS—DVM with 4 years’ experience and validated by MW—DVM, Ph.D., dr hab. prof. UPWr, diplECVN.

### 2.2. MR Protocol

All MR examinations were performed in the magnetic resonance imaging laboratory of the Center of Experimental Diagnostics and Innovative Biomedical Technologies at Wrocław University of Environmental and Life Sciences.

Before MRI examinations all dogs were anaesthetized with the same protocol. Sedation was performed with medetomidine (Cepetor^®^, CP-Pharma, Burgdorf, Germany) 0.1 mL/10 kg and butorphanol (Torbugersic^®^, Zoetis, Warsaw, Poland) 0.1 mL/10 kg administered intramuscularly. General anaesthesia induction was performed with propofol (Propofol Lipuro^®^ B Braun, Melsungen AG, Meldungen, Germany) 1 mg/1 kg intravenously. All dogs were intubated and inhalation anaesthesia was applied with isoflurane. The vapor setting was 3–4% at induction with oxygen flow at 60 mL/kg/min; after 5 min, it was reduced to 1.5–3% for maintenance with oxygen flow at 20 mL/kg/min.

All MRI studies were performed on a 1.5 T MR scanner (Philips Ingenia, Philips Healthcare, Eindhoven, Holland). An abbreviated MRI protocol created for CM-SM syndrome screening was used. If required, the T1, post-contrast T1 and FLAIR sequences were additionally performed. The standard MRI protocol included transverse T2-weighted images (TR 5330.1 ms/TE 120 ms, slice thickness 2.5 mm, FOV 135 × 135 mm, matrix size 272 × 161, voxel 0.5 × 0.72 mm, slice thickness 2.5 mm, slice gap-0.2 mm, NSA-4) and sagittal T2-weighted images (TR4623.9 ms/TE110 ms, slice thickness 2 mm, FOV 175 × 410 mm, matrix size 252 × 424, voxel 0.7 × 0.9 mm, slice thickness 2 mm, slice gap-0.2 mm, NSA-5). MRI examination covered the brain and the whole spinal cord of each subject.

All standard structural MRI sequences (transverse and sagittal T2) of the spinal cord of all dogs were assessed and graded by the authors (AB, KOS) and validated by a diplomate veterinary neurologist (MW) using the standardized British Veterinary Association (BVA) scale [16,17,18,19]. Syringomyelia was graded as follows: Grade 0: normal (no central canal dilation, no presyrinx, no syrinx), Grade 1: central canal dilation (CCD) <2 mm in diameter, and Grade 2: syringomyelia (central canal dilation which has an internal diameter ≥2 mm, separate syrinx or pre-syrinx with or without central canal dilation) [11]. Based on these findings, dogs were divided into 2 groups: without MRI signs of SM (non-SM) and a study group with visible syringomyelia (SM) (Figure 1). The MR images were evaluated by the authors (A.E.B., K.O.-S.–and validated by diplomate neurologist MW and human radiologist, an MRI expert at Medical University AZ).

### 2.3. DTI Protocol and Postprocessing

The DTI sequence used a single-shot, interleaved, multi-slice, spin echo, echo-planar acquisition performed in the sagittal plane using the following scan parameters: TR 5653 msec; TE 116 msec; field-of-view 160 × 160 mm; slice thickness 1.5 mm; imaging matrix 108 × 105 (in-plane resolution: 1.5 × 1.5 mm), slice gap-0 mm and NSA-3. A total of 46–50 slices were acquired. DTI was measured with an average directional resolution, i.e., in 15 diffusion directions with a b-factor of 800. The total acquisition time was 17 min and 54 s. The SENSE factor was set between 1.5 and 3.0. A balanced pair of diffusion gradients were used to minimize eddy current artifacts. No respiratory or cardiac gating was performed. A sagittal DTI acquisition plane was chosen to maximize anatomical coverage of the cervical spinal cord while keeping the acquisition time as short as possible.

Post-processing of the DTI data was performed using Philips DTI Fiber Trak Software. The reconstruction of white matter tracts was performed by manually drawing the region of interest (in linear shape) from the occipital bone to the fourth cervical vertebrae on the sagittal plane. ADC and FA metrics were measured on ADC and FA maps using manual placement of 3 regions of interest (ROIs) in the centre of the cervical spinal cord in the midsagittal plane at the level of 3 intervertebral discs between C1 and C4 (ROI-1 at C1-C2, ROI-2 at C2-C3, ROI-3 at C3-C4); ROIs were linear in shape and similar in size. In the study group, ROI-1 was always applied on the spinal cord without visible pathological changes in standard structural MRI protocol, while ROIs 2 and 3 included the syrinx (Figure 2).

The correct positioning of each ROI was cross-checked on all available structural and DTI images. Care was taken to avoid partial volume effects with adjacent CSF or bony structures. FA and ADC metrics were calculated and averaged over the selected voxels for each ROI. ADC and FA values were not measured within the thoracic spinal cord because of the greater susceptibility to artifacts in these regions [23].

### 2.4. Statistical Analysis

Statistical analyses were performed using commercial software system Statistica 13.3 (TIBCO Software Inc., Palo Alto, CA, USA, 2017).

Testing for the normal distribution of the data obtained was performed with the Shapiro–Wilk normality test. The assumptions about the normality of the data distribution were maintained in the analysed cases.

The Student’s t-test, for independent samples, was employed to compare age, body weight, and DTI parameters (ADC and FA values) between the non-SM and SM groups. One-way analysis of variance (ANOVA) was used to compare the differences between the average FA and ADC values between ROI-1, ROI-2, and ROI-3 in the non-SM (control) group, SM-asymptomatic and SM-symptomatic groups. If the null hypothesis of equality of all means was rejected, the post hoc cross-comparison test by Tukey’s HSD method was used. In all analyses, *p* < 0.05 was considered to be statistically significant.

## 3. Results

Out of the 72 Cavalier King Charles Spaniel dogs admitted to the Department of Internal Disease with Clinic for Horses, Dogs and Cats, the final number of dogs enrolled in the study was 40 (age 6–70 months, weight 3.5–9.5 kg, 11 males, 19 females), 8 of these 40 were symptomatic and 32 asymptomatic. MRI confirmed SM in 18 dogs and 12 dogs did not have pathological findings (non-SM group—control group). In the SM group, eight dogs were found to have clinical symptoms (SM-symptomatic) and 10 dogs had no clinical symptoms (SM-asymptomatic). Eight dogs out of 10 in the SM group were symptomatic (SM-symptomatic group), and showed neurological deficits (5/8 showed weakened flexor reflex on front limbs, delayed proprioception, neck pain and 3/8 showed weakened flexor reflex on front limbs, neck pain and delayed proprioception).

There were no significant differences in age and body weight between the non-SM (control group) and SM dogs as well as between symptomatic and asymptomatic groups.

None of the animals in the non-SM group showed clinical symptoms of CM-SM syndrome. Within the SM group, there were 10 asymptomatic and 8 symptomatic dogs. In the SM-asymptomatic group, three dogs were assessed with Grade 1 syringomyelia, and seven with Grade 2 (median Grade 1.7). In four of them, a syrinx was located in the cervical segment of the spinal cord (in one C2-C4, in one C2-C5, in one C2-C6, and one C1-C6), in one it extended to Th1 and in five of them a syrinx extended from cervical to the lumbar spinal cord (in 4 C2-L3, in one C2-L4). Eight dogs out of ten in the SM group were symptomatic (SM-symptomatic group), and showed neurological deficits (weakened flexor reflex on front limbs and delayed proprioception). In 2/8 dogs a syrinx was located in the cervical segment of the spinal cord (in one C1-C3, in one C1-C6); in six dogs, a syrinx extended from the cervical to lumbar spinal cord (in four of them C2-L3, and in two of them C2-L4). In 2/8 dogs, Grade 1 SM was found and in six the diameter of the syrinx exceeded 2 mm (Grade 2) (median Grade 1.75).

There were no visible spinal cord changes in characteristics for so-called CM-SM syndrome in the non-SM group.

Statistical analysis showed no significant differences in FA and ADC values between the SM and non-SM groups. However, a tendency for increase of ADC values and decrease of FA values in all three ROIs were found in the SM compared to the non-SM. The comparison of ADC and FA values between the SM and non-SM groups is presented in Table 1, Table 2, Table 3 and Table 4, and Figure 3 and Figure 4.

Statistically significant differences between non-SM and SM-symptomatic groups, and also between SM-asymptomatic and SM-symptomatic groups were found.

In the SM-symptomatic group compared to the SM-asymptomatic group, decreased FA values were observed in all three ROIs, but in ROI-2 the differences were not statistically significant. In the SM-symptomatic group compared to the SM-asymptomatic group, increased ADC values were observed and the differences were statistically significant. The comparison of ADC and FA values between the symptomatic (eight dogs) and asymptomatic (ten dogs) groups is presented in Table 5 and Figure 5 and Figure 6.

Compared to non-SM, SM-symptomatic dogs showed decreased FA values in ROI-1 (*p* = 0.000022); and ROI-3 (0.000019) also increased ADC values in ROI-1 (*p* = 0.00002), ROI-2 (*p* = 0.002627) and ROI-3 (0.000003).

Compared to SM-asymptomatic, SM-symptomatic dogs showed decreased FA values in ROI-1 (0.000018) and ROI-3 (0.000019), and also increased ADC values in ROI-1 (0.000024), ROI-2 (0.003334) and ROI-3 (0.000001).

## 4. Discussion

Due to the discrepancy described in the literature between the severity of clinical symptoms of CM-MS syndrome and structural changes visible on standard MRI of the spinal cord, an objective DTI sequence was used in this study [1,2,3,4,5,6,7,8,9,10,11,24,25]. We hypothesized that DTI would be a good candidate for prediction regarding the prevalence of the symptomatic CM-SM course and that this would be of great value in canine and human patients.

In this study, we found statistically significant differences in all three ROIs in ADC values and in ROI-1 and ROI-3 in FA values between SM-asymptomatic and SM-symptomatic animals. We assumed that the presence of CM-SM-specific clinical symptoms but without imaging visible SM of the spinal cord, may indicate the presence of potential structural changes in the spine itself, which, however, are not detectable during routine MR imaging but can be detected by using the DTI technique.

There was an observed tendency towards reduced values of FA and increased values of ADC in dogs with SM, but they were not statistically significant.

Fractional anisotropy is a parameter considered to be a marker of fibre tract integrity while an increase in ADC values suggests a demyelinating process [21,22,24,26,27]. In agreement with our hypothesis, changes in both DTI parameters confirmed disturbed integrity and organization of the spinal cord in SM-symptomatic dogs.

Based on studies on syringomyelia in human medicine and information regarding this disease we expected that in patients with the CM-SM syndrome, FA values would decrease while ADC measurements should increase at the syrinx level [6,7,8,9,10,11,28,29,30,31,32]. In human DTI studies regarding CM-SM syndrome in 23 patients, a positive correlation between FA values and microstructural damage in the course of SM was found at the syrinx level [6,7,8,9,10,11]. Our results are also in agreement with research by Yan et al., Hatem et al. and Wu et al. who also found similar DTI outcomes in both SM and non-SM groups in humans. In the group tested at the syrinx level, the FA values decreased in both humans and animals [6,7,8,9,10,11]. This suggests that CKCS dogs may be a valuable model for the study of human syringomyelia.

An important aspect investigated in the work of Yan et al. was a comparison of symptomatic and asymptomatic human patients. The researchers observed that FA values were lower in symptomatic patients than in asymptomatic patients. This finding corresponds to the results of our study. Because there was an observed tendency towards reduced values of FA and increased values of ADC in dogs with SM, both parameters should be taken into consideration when assessing microstructural damage in non-SM but symptomatic dogs. Further research in this area on a larger group of canine patients will be continued, as the obtained results suggest that a relationship between DTI parameters and structural damage in this group of patients exists. Confirmation of this correspondence would potentially facilitate the diagnosis and further prognosis in patients with CM-SM syndrome.

DTI values can be used by breeders for preventive purposes in young dogs. They would be aimed at assessing whether an individual has the predisposition to develop symptoms of the syndrome and, on this basis, decide whether an individual is suitable for breeding.

It is worth noting that syringomyelia is a disorder related primarily to the grey matter, while the FA and ADC parameters are believed to better reflect white matter integrity. Earlier studies in human medicine have shown that FA and ADC parameters are altered in patients with CM-SM. This suggests that grey matter changes had influenced the development of functional disorder in white matter of the spinal cord. Nevertheless, the group of patients were not large enough to show a clear correlation between clinical symptoms and FA values [6,10].

In the study of Yan et al., FA measurements were performed caudally to the syrinx [6]. The researchers observed a reduction in FA values in patients in the SM group compared to the non-SM group. However, these data cannot be considered statistically significant and should be interpreted carefully. In our study, owing to the different size of the syrinx in each patient, we decided not to perform such measurements as they may not have been reliable, and this fact could have influenced the results of the DTI parameters. In small animals, it is also more challenging because of the small size of the spinal cord especially in the thoracic level. We believe that evaluation of FA and ADC values below the syrinx may be of great importance and may generate very interesting information about the rate of degeneration below the lesion level in a normal appearing spinal cord. Results from these studies that correlate with the severity of clinical symptoms could be used in the future for an objective assessment of the rate of spinal cord damage and prognosis for individual patients.

A beneficial addition to this research would be to supplement it with a non-SM symptomatic group. Hypothetically, we would expect a decrease in the FA values and an increase in the ADC values in all three ROIs. The value of both parameters would be interesting when the radiographic changes are invisible and clinical symptoms appear.

## 5. Study Limitations

The principal limitation of this study was the small number of dogs in each study group. Although the study group allowed the evaluation of a comparison between FA or ADC values and the severity of clinical symptoms, it did not allow findings of statistically significant differences between non-SM and SM dogs. Future prospective studies should incorporate larger data groups to increase the statistical power of the comparisons between standard protocol MRI results and DTI metrics. In clinical practice, we observed non-MS symptomatic dogs that were not observed in the currently presented groups. As DTI is a sensitive technique, we expect a lowering of the FA or/and an increase in ADC parameters in such cases, this assumption however has to be confirmed in a clinical set.

Another limiting factor in our study was the spatial resolution, which did not allow for differentiation between grey and white matter within the spinal cord; therefore, the ROI analysis included regions containing both grey and white matter. The same methodology has been already used in the assessment of human patients [6,23]. It is worth bearing in mind that this methodology was used in paediatric and infant patients, whose spinal diameter is much smaller than in adult humans [23]. Therefore, we believe that despite the small diameter of the spinal cord, this method can be successfully used in this size of the veterinary patient. A point of note is that the measurements were made without excluding the syrinx. Taking measurements without the syrinx was not possible with our software, either in the spatial or transverse plane, taking into account the small size of a spinal cord.

Another limitation was the cross-sectional character of the study. It would be interesting to perform a longitudinal study and continue follow-up of dogs with SM and no symptoms to investigate if FA and ADC values could be of predictive value in anticipation of the clinical course in these patients.

Additionally, it should be emphasized that a dog’s spinal cord has a small diameter which makes DTI acquisition very challenging. We limited our study to the cervical part which is the largest portion of the spinal cord and also because artifacts from the heart and major vessels are less pronounced at this level compared to the thoracic or lumbar spinal cord. However, the measurements in our study were repeatable, therefore we believe that this is an appropriate method for the evaluation of CM-SM in animals.

Finally, the studied DTI sequence sampled only one single high b-value along each diffusion-encoding gradient. A more detailed characterization of the diffusion decay over time (mono-exponential versus multi-exponential) requires the use of multiple b-values.

Lastly, advanced cardiac and respiratory gating methods were not performed in our research. Therefore, CSF pulsations and respiratory-related spinal cord motion may have impacted the quality of the DTI data. Images were, however, carefully evaluated for image distortion, motion artifacts and field homogeneity.

## 6. Conclusions

The results obtained in our research correlate with the results of studies conducted in human medicine which showed decreased FA values and increased ADC values in spinal cord diseases leading to disorders of nervous tissue integrity [6]. This means that CKCS dogs with CM-SM syndrome can potentially provide an excellent model for studying CM-SM in humans, and in particular that CM-SM syndrome occurs in this dog breed spontaneously, which is a contributory advantage for animal model research [33,34].

The use of DTI in the MRI evaluation of CM-SM patients carries a potential value in the development of a clinically useful protocol for an objective assessment of the spinal cord and to understand what processes lie at the basis of many diseases, the diagnosis of which is currently difficult. Tests performed on animals may contribute to progress in the diagnosis of CM-SM in humans [35,36,37]. We believe that in the future DTI could be a helpful technique in the study of syringomyelia in CKCS and for establishing a prognosis based on the development of clinical symptoms in young dogs.

## Figures and Tables

**Figure 1 animals-12-03405-f001:**
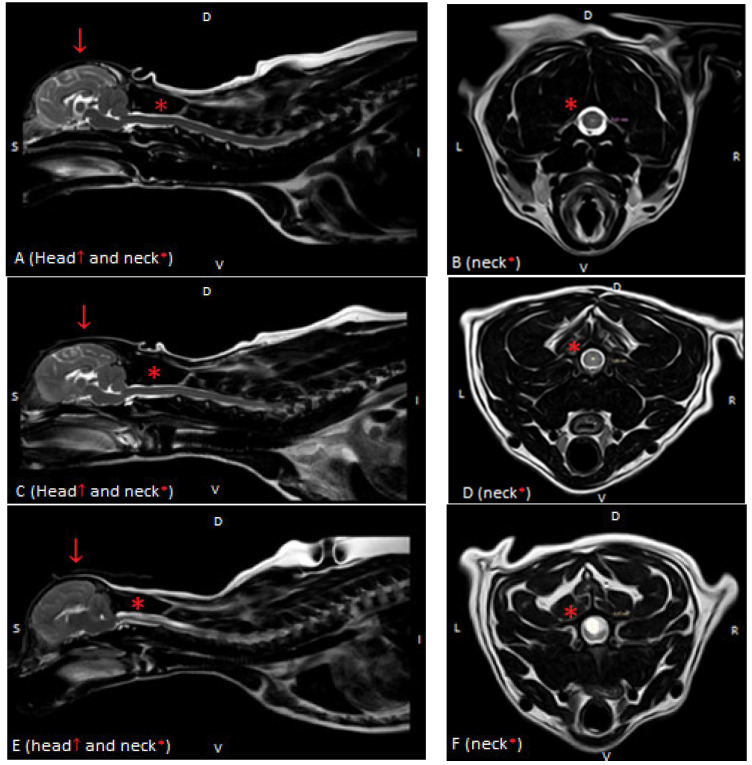
T2-weighted MR images of the spinal cord in the sagittal (**A**,**C**,**E**) and transverse (**B**,**D**,**F**) planes in three dogs showing: (**A**,**B**)—normal spinal cord (Grade 0), (**C**,**D**)—central canal dilatation (Grade 1) and (**E**,**F**)—syringomyelia (Grade 2). There is severe flattening of the caudal fossa with marked compression of the cerebellum and crowding of the cerebellum (**A**,**C**,**E**).

**Figure 2 animals-12-03405-f002:**
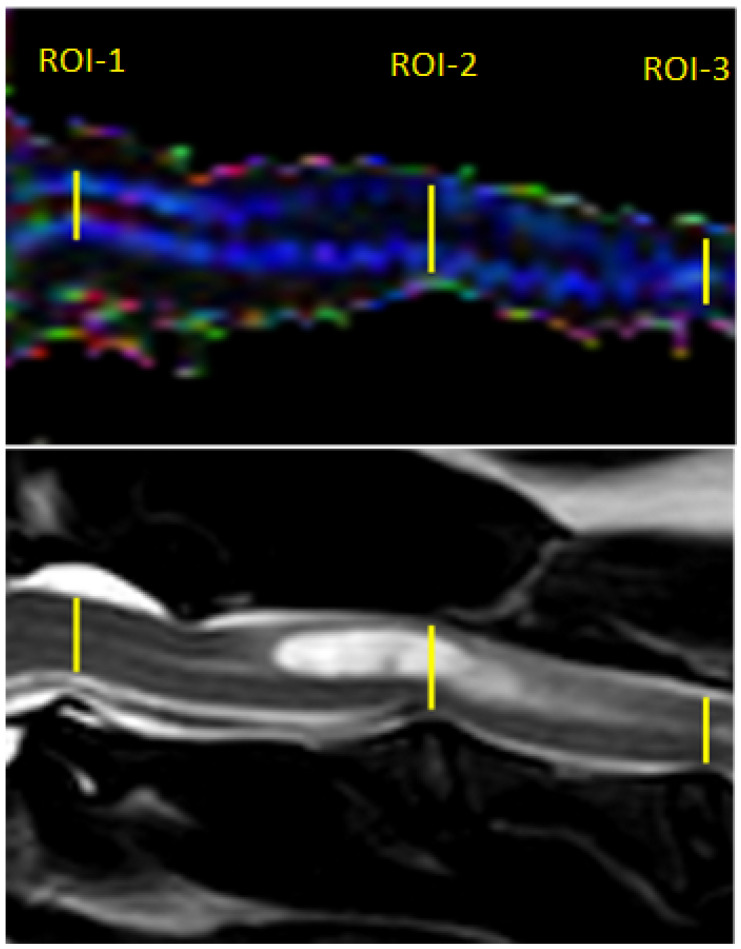
Sagittal T2-weighted MR image and FA map of the spinal cord showing ROIs placement within the spinal cord of the SM dog.

**Figure 3 animals-12-03405-f003:**
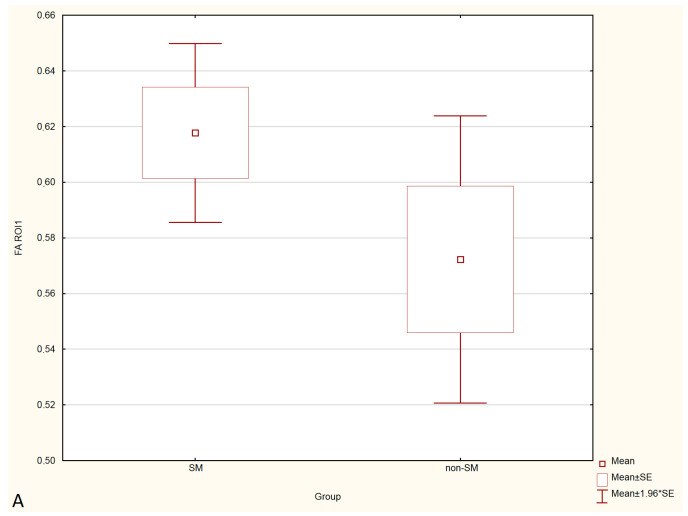
Boxplots comparing FA values in ROI-1 (**A**), ROI-2 (**B**), ROI-3 (**C**) between the SM and non-SM groups.

**Figure 4 animals-12-03405-f004:**
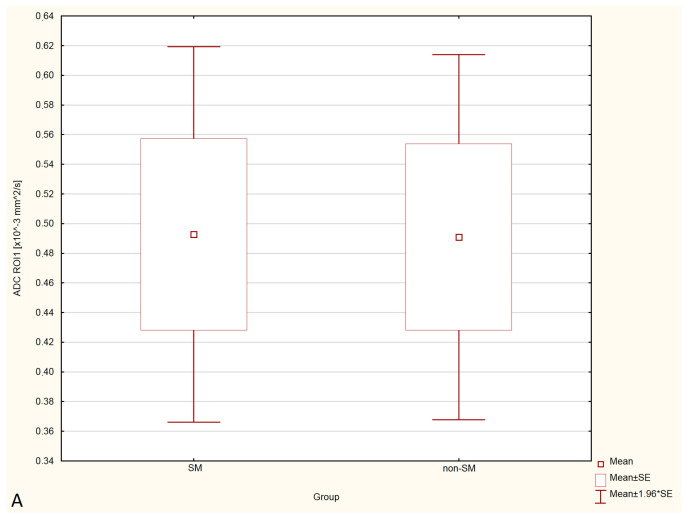
Boxplots comparing ADC values in ROI-1 (**A**), ROI-2 (**B**), ROI-3 (**C**) between the SM and non-SM groups.

**Figure 5 animals-12-03405-f005:**
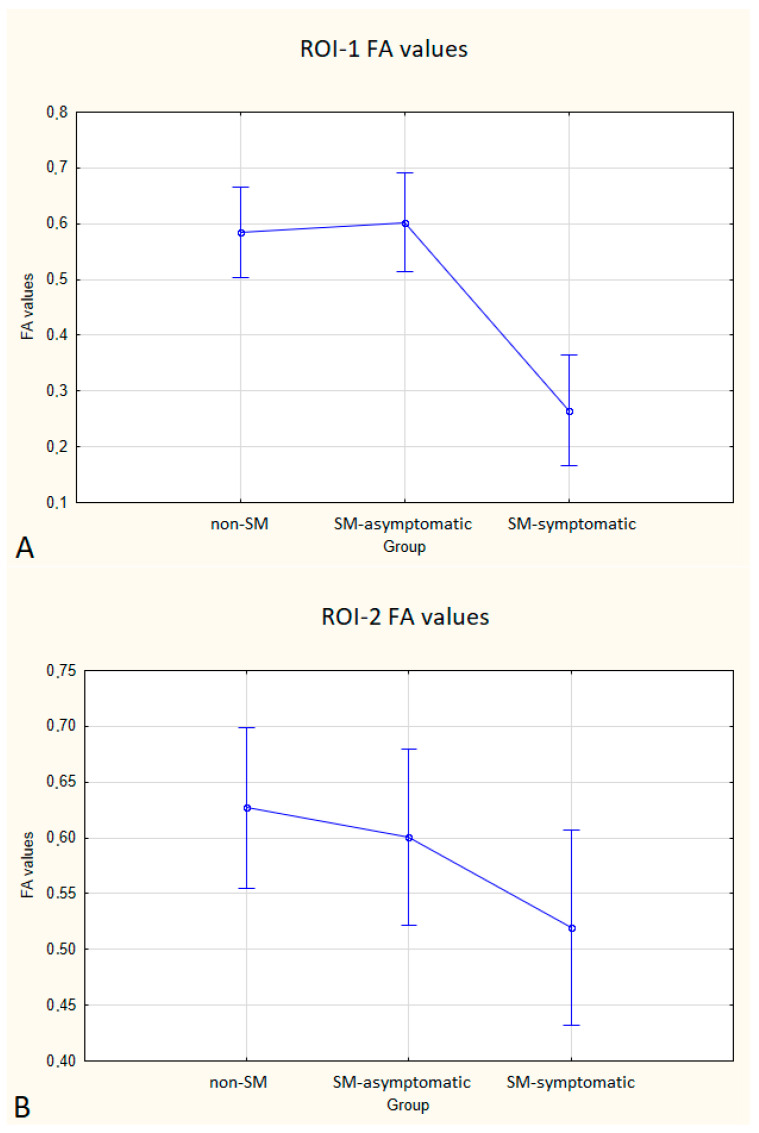
Boxplots comparing FA values in ROI-1 (**A**), ROI-2 (**B**), ROI-3 (**C**) between the non-SM, SM-symptomatic and SM-asymptomatic groups.

**Figure 6 animals-12-03405-f006:**
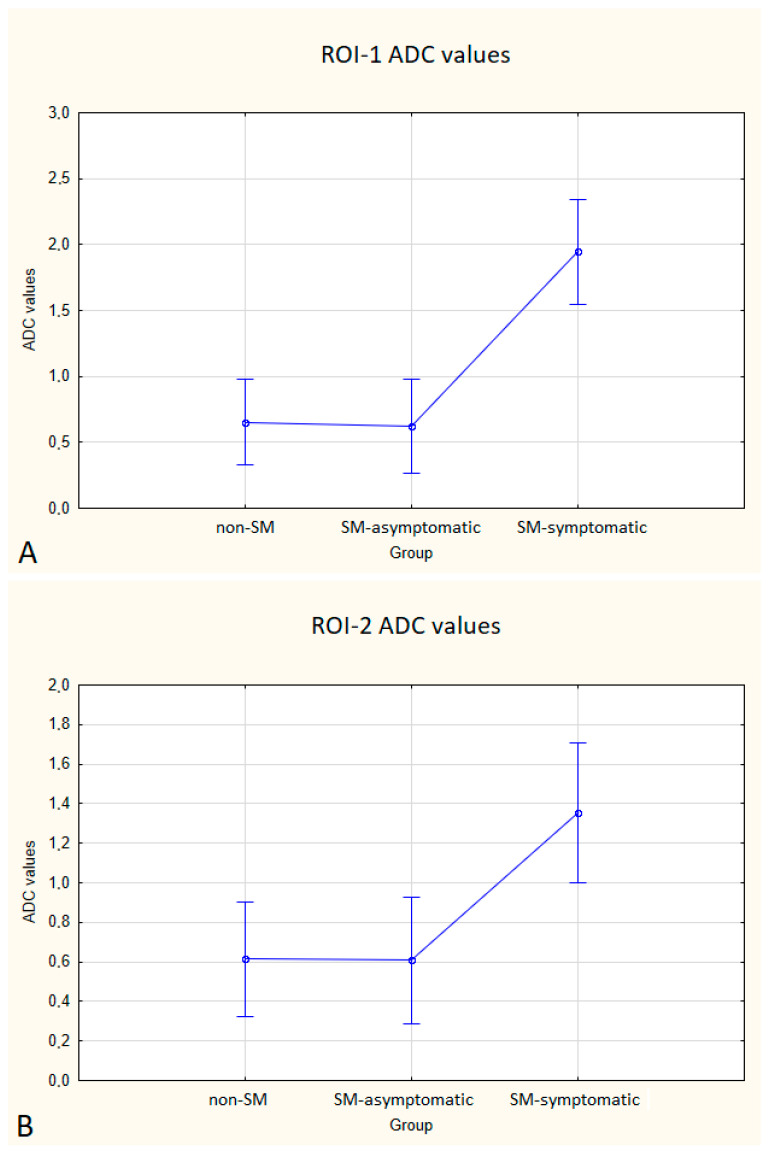
Boxplots comparing ADC values in ROI-1 (**A**), ROI-2 (**B**), ROI-3 (**C**) between the non-SM, SM-symptomatic and SM-asymptomatic groups.

**Table 1 animals-12-03405-t001:** Clinical and MRI results of the non-SM group with FA and ADC measurements from ROI-1, ROI-2 and ROI-3.

Non-SM GROUP									
Patient	Clinical Symptoms (Yes/No)	MRI Results		ROI-1		ROI-2		ROI-3	
		Location	Grade	FA Value	ADC Value	FA Value	ADC Value	FA Value	ADC Value
1.	No	-	0	0.673	0.478	0.608	0.684	0.643	0.516
2.	No	-	0	0.466	0.228	0.747	0.161	0.669	0.13
3.	No	-	0	0.634	0.545	0.672	0.334	0.724	0.165
4.	No	-	0	0.468	0.851	0.506	0.149	0.626	0416
5.	No	-	0	0.565	0.546	0.595	0.524	0.602	0.329
6.	No	-	0	0.592	0.379	0.651	0.347	0.56	0.421
7.	No	-	0	0.562	0.446	0.63	0.326	0.544	0.32
8.	No	-	0	0.618	0.454	0.618	0.415	0.654	0.277
9.	No	-	0	0.663	1.013	0.628	1.221	0.716	0.712
10.	No	-	0	0.374	1.781	0.537	1.024	0.381	1.319
11.	No	-	0	0.559	0.465	0.645	1.08	0.683	0.931
12.	No	-	0	0.839	0.646	0.685	1.104	0.569	0.579
			SD	0.119	0.412	0.063	0.393	0.093	0.341

FA—fractional anisotropy, ADC—apparent diffusion coefficient, ROI—region of interest.

**Table 2 animals-12-03405-t002:** Clinical and MRI results of the SM-asymptomatic group with FA and ADC measurements from ROI-1, ROI-2 and ROI-3.

SM-Asymptomatic GROUP									
Patient	Clinical Symptoms (Yes/No)	MRI Results		ROI-1		ROI-2		ROI-3	
		Location	Grade	FA Value	ADC Value	FA Value	ADC Value	FA Value	ADC Value
1.	No	C2-C4	1	0.635	0.566	0.591	0.486	0.605	0.486
2.	No	C2-Th1	1	0.613	0.474	0.738	0.202	0.671	0.183
3.	No	C2-C5	2	0.62	0.349	0.623	0.342	0.603	0.313
4.	No	C2-C6	2	0.563	0.508	0.655	0.303	0.618	0.395
5.	No	C2-L3	2	0.671	0.351	0.655	0.541	0.666	0.318
6.	No	C2-L3	2	0.69	0.233	0.625	0.194	0.527	0.395
7.	No	C2-L3	2	0.571	0.673	0.486	0.616	0.621	0.208
8.	No	C1-L3	1	0.695	1.15	0.729	0.885	0.649	0.398
9.	No	C1-C6	2	0.386	1.092	0.384	1.783	0.746	0.718
10.	No	C2-L4	2	0.579	0.788	0.522	0.718	0.529	0.577
			SD	0.089	0.31	0.11	0.47	0.065	0.162

FA—fractional anisotropy, ADC—apparent diffusion coefficient, ROI—region of interest, C—cervical spinal cord, Th—thoracic spinal cord, L—lumbar spinal cord.

**Table 3 animals-12-03405-t003:** Clinical and MRI results of the SM-symptomatic group with FA and ADC measurements from ROI-1, ROI-2 and ROI-3.

SM-Symptomatic GROUP									
Patient	Clinical Symptoms (Yes/No)	MRI Results		ROI-1		ROI-2		ROI-3	
		Location	Grade	FA Value	ADC Value	FA Value	ADC Value	FA Value	ADC Value
1	Yes	C3-L3	2	0.179	2.938	0.531	1.335	0.5	1.402
2.	Yes	C1-C3	1	0.436	1.265	0.714	0.984	0.278	1.656
3.	Yes	C2-L4	2	0.111	2.667	0.627	1.034	0.21	2.118
4.	Yes	C2-L3	2	0.098	2.711	0.447	1.62	0.576	0.785
5.	Yes	C1-C6	1	0.684	0.362	0.607	0.782	0.362	0.946
6.	Yes	C2-L4	2	0.188	2.052	0.192	2.744	0.297	2.537
7.	Yes	C2-L3	2	0.236	1.396	0.327	1.42	0.475	1.496
8.	Yes	C2-L3	2	0.189	2.178	0.711	0.912	0.417	1.249
			SD	0.199	0.880	0.186	0.629	0.124	0.582

**Table 4 animals-12-03405-t004:** Comparison of FA and ADC values in ROI-1, ROI-2, ROI-3 between the SM and non-SM group.

	Mean FA Values			Mean ADC Values		
	SM Group	Non-SM Group	*p*-Value	SM Group	Non-SM Group	*p*-Value
ROI-1	0.452	0.584	0.07	1.209	0.653	0.06
ROI-2	0.565	0.627	0.19	0.939	0.614	0.13
ROI-3	0.614	0.519	0.06	0.899	0.510	0.08

FA—fractional anisotropy, ADC—apparent diffusion coefficient, ROI—region of interest.

**Table 5 animals-12-03405-t005:** Comparison of FA and ADC values in ROI-1, ROI-2, ROI-3 between the non-SM, SM-symptomatic and SM-asymptomatic groups.

	Mean FA Values			Mean ADC Values		
	SM-Symptomatic Group	SM-Asymptomatic Group	*p*-Value	SM-Symptomatic Group	SM-Asymptomatic Group	*p*-Value
ROI-1	0.265	0.602	0.000170	1.956	0.618	0.00002
ROI-2	0.520	0.601	0.16250	1.354	0.607	0.003334
ROI-3	0.389	0.624	0.000001	1.524	0.399	0.000001

FA—fractional anisotropy, ADC—apparent diffusion coefficient, ROI—region of interest.

## Data Availability

The data presented in this study are available on request from the corresponding author.

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
