# Peer review of "Diffusion Tensor Imaging in Syringomyelia Secondary to Chiari Malformation in Cavalier King Charles Spaniel—A Preliminary Study"

_animals, 2022, doi:10.3390/ani12233405_

Round 1

Reviewer 1 Report

Dear authors,

thank you for this interesting preliminary paper on the use of diffusion tensor imaging to assess damage to the spinal cord in CCKS affected by syringomyelia. It would be a nice addition to the literature on the subject and it looks like a promising tool to add information about syringmyelia.
However, there are some parts of the paper that needs to be revised, are missing or are unclear.
Please find below some general comments and some that I would like you to address.

Across the whole paper you use the term radiologically or radiological when referring to the imaging side of things; however, since the paper is about MR this term is incorrect; please substitute it everywhere with "imaging results" or "MR results" or similar more appropriate terms.( e.g. lines 21,62, 68, 196, 489 etc).

Please revise the use of English language in certain sentences:
E.g.Line 73 : "the severity....is not commensurate"

The title is appropriate and describes the aim of the paper

In the summary, from lines 13 to 16 you are trying to make two points, but they look very similar; please rewrite it in a clearer way.

Abstract:
- There is no need to give so many details about the patients in the abstract (weight etc)line 36 to 38. IN general, sentences from line 36 to 44 should be written less as results and more in an abstract format, less specific and more to the point. 

Introduction:
- Line 70 to 76, like in the summary there are three sentences which are very similar about severity of the disease and harmfulness of the disease. Please rewrite your concepts in a clearer way, as it appears like you are just repeating the same concept three times.
- The reason for using diffusion tensor imaging are clearly explained. It would be nice to very briefly (and as clearly as possible) also explain here what is diffusion tensor imaging.

MM
-The type of study should be stated at the beginning of the paragraph (eg. prospective, control etc)
- The inclusion and exclusion criteria should be listed: were all CCKS admitted to the hospital with neuro symptoms in those 2 years  included in the study? Which ones were excluded and why?Etc.
- Lines 117 to 122 belong in the results section, not in the MM
- How is it possible that no ethical approval was needed in a study whereby animals need to be anaesthetised?Did you apply for exemption? If so, it should say so, not that ethical approval was not needed.
- Was the severity of  the clinical symptoms graded in any way? By who?
- The qualification and level of experience of the people performing the neuro exam should be mentioned here (line 126-7)
- Please mention the qualifications/level of experience of the people grading the MR images (Line 155)
- Also very important: were the people grading the MR images blind to the group of pertenance of the dogs being assessed?
- Fig. 1 is good. It would be nice to add that in C there is severe flattening of the caudal fossa with marked compression of the cerebellum and crowding of the cerebellum (Also in A and B)
- Fig. 2, please flip in a standard horizontal position of the spinal cord.

Results:
-The results sections should start in a similar way to this: out of the TOT number of dogs admitted to the hospital.....the final number of dogs enrolled in the study were TOT. .of these TOT were symptomatic etc etc. Please edit accordingly.
- Here is also were age, sex, weight etc should be mentioned for the total group of dogs included.
- Please look at use of language in line 240, this sentence is not clear.
- Across the paper it is not clear whether results were statistically significant or not: e.g. :lines 302-3 of the results vs lines 48-9 of the abstract. These sentences appear to contradict one another.

Discussion
- On line 370 what is meant by " an objective MRI sequence was used"?
- I am confused by what stated in lines 382-3 and then 403, as they seem to contradict what mentioned in the abstract in lines 48-9. Could you please find a way to express these findings in a clearer way? Would it be correct to say that there was a tendency towards reduced values of FA and increased values of ADC in dogs with SM and in symptomatic dogs, but they were not statistically significant?

References are relevant and mostly recent. However, I was able to find two references which are very recent (both 2021)and relevant and that are not included. 

1. Clinical Application of Diffusion Tensor Imaging in Chiari Malformation Type Ie Advances and Perspectives. A Systematic Review
Lukasz Antkowiak, Marta Rogalska, Piotr Stogowski,Karolina Anuszkiewicz,Marek Mandera

2. Specific microstructural changes of the cervical spinal cord in syringomyelia estimated by diffusion tensor imaging

Author Response

Reviewer 1

Please find below some general comments and some that I would like you to address.

Across the whole paper you use the term radiologically or radiological when referring to the imaging side of things; however, since the paper is about MR this term is incorrect; please substitute it everywhere with "imaging results" or "MR results" or similar more appropriate terms.( e.g. lines 21,62, 68, 196, 489 etc).

Dear Reviewer,

Thank You for all Your comments and advices. The sentences we’ve changed in manuscript are in green colour.

Please revise the use of English language in certain sentences:

E.g.Line 73 : "the severity....is not commensurate"

We changed it in manuscript.

In the summary, from lines 13 to 16 you are trying to make two points, but they look very similar; please rewrite it in a clearer way.

As above

Abstract:

- There is no need to give so many details about the patients in the abstract (weight etc)line 36 to 38. IN general, sentences from line 36 to 44 should be written less as results and more in an abstract format, less specific and more to the point.

Dear Reviewer,

You asked us to make our abstract more specific and we are asked by a Reviever 3 to add some numerical values in our abstract. We tried to find a compromiss and we hope it will work.

Introduction:

- Line 70 to 76, like in the summary there are three sentences which are very similar about severity of the disease and harmfulness of the disease. Please rewrite your concepts in a clearer way, as it appears like you are just repeating the same concept three times.

It is changed in manuscript

- The reason for using diffusion tensor imaging are clearly explained. It would be nice to very briefly (and as clearly as possible) also explain here what is diffusion tensor imaging.

As above

MM

-The type of study should be stated at the beginning of the paragraph (eg. prospective, control etc)

As above

- The inclusion and exclusion criteria should be listed: were all CCKS admitted to the hospital with neuro symptoms in those 2 years  included in the study? Which ones were excluded and why?Etc.

Ear diseases, dermatological, orthopaedic problems, infection or other non-neurological problems were ruled out in all dogs enrolled in this study.

We also add it in manuscript

- Lines 117 to 122 belong in the results section, not in the MM

All animals underwent the same study protocol that included a detailed clinical and neurological examination followed by MRI examination. – we add predefined in manucript and left it in M&M

- How is it possible that no ethical approval was needed in a study whereby animals need to be anaesthetised?Did you apply for exemption? If so, it should say so, not that ethical approval was not needed.

All studies were performed in patients, that were submitted for the consultation by owners to the Veterinary Clinic if the University. This does not require an approval of the Local Ethics Committee as no experimental animals were used in this study. The research (as prospectively planned but post-processing study) was performed on the images and data obtained from clinical trials.

We’ve also a e-mail conversation about it and Editor wrote us that in this case we don’t need Local Ethics Commitee Approval. We are asked to send owners permission and of course we will send it.

- Was the severity of  the clinical symptoms graded in any way? By who?

The severity of clinical symptoms was assessed during a neurological examination by AB, KOS, MW. However, in these studies we did not focus on the correlation between the severity of clinical symptoms, so it was not graded. The goal was to determine whether the correlation exists at all. Thank You very much for this comment. We believe that it is very interesting to conduct studies taking into account the severity of clinical signs in the future, but on larger group of animals.

- The qualification and level of experience of the people performing the neuro exam should be mentioned here (line 126-7)

mentioned here (line 126-7)

Than you, we added qualification of the researches.

- Please mention the qualifications/level of experience of the people grading the MR images (Line 155)

As above

- Also very important: were the people grading the MR images blind to the group of pertenance of the dogs being assessed?

Those who evaluated the MRI scans also performed the neurological examination. However during the evaluation of MRI study, the evaluators did not have access to the results of the neurological examination. DTI is a method that allows an objective assessment of the spinal cord because it presents numerical values. MRI and DTI assessment was performed blindly. Therefore in our opinion, it was sufficient.

- Fig. 1 is good. It would be nice to add that in C there is severe flattening of the caudal fossa with marked compression of the cerebellum and crowding of the cerebellum (Also in A and B)

- Fig. 2, please flip in a standard horizontal position of the spinal cord.

We changed our Figures according to Reviewers advices.

Results:

-The results sections should start in a similar way to this: out of the TOT number of dogs admitted to the hospital.....the final number of dogs enrolled in the study were TOT. .of these TOT were symptomatic etc etc. Please edit accordingly.

As above

- Here is also were age, sex, weight etc should be mentioned for the total group of dogs included.

As above

- Please look at use of language in line 240, this sentence is not clear.

However, a tendency for increase of ADC values and decrease of FA values in all three ROI-s were found in the SM compared to the non-SM

As above

- Across the paper it is not clear whether results were statistically significant or not: e.g. :lines 302-3 of the results vs lines 48-9 of the abstract. These sentences appear to contradict one another.

We found statistically significant differences in ADC values in all three ROIs and in FA values in ROI-1 and ROI-3 between non-SM, SM-asymptomatic and SM-symptomatic groups.

No statistically significant differences in FA and ADC values between the SM and nonSM groups were found.

We’ve paraphrased lines 302-303 to be more clear.

Discussion

- On line 370 what is meant by " an objective MRI sequence was used"?

We meant that the DTI sequence is an objective MRI sequence. We’ve paraphrase this sentence because it could be unclear.

- I am confused by what stated in lines 382-3 and then 403, as they seem to contradict what mentioned in the abstract in lines 48-9. Could you please find a way to express these findings in a clearer way? Would it be correct to say that there was a tendency towards reduced values of FA and increased values of ADC in dogs with SM and in symptomatic dogs, but they were not statistically significant?

As above

References are relevant and mostly recent. However, I was able to find two references which are very recent (both 2021)and relevant and that are not included.

  1. Clinical Application of Diffusion Tensor Imaging in Chiari Malformation Type Ie Advances and Perspectives. A Systematic Review

Lukasz Antkowiak, Marta Rogalska, Piotr Stogowski,Karolina Anuszkiewicz,Marek Mandera

  1. Specific microstructural changes of the cervical spinal cord in syringomyelia estimated by diffusion tensor imaging

Weifei Wu,

 Xiangxiang Li,

 Zong Yang,

 Neng Ru,

 Fan Zhang,

 Jie Liang &

 Ke Zhang

Thank You. We add more references

Reviewer 2 Report

Wrzosek et al utilized diffusion tensor imaging (DTI) in Cavalier King Charles Spaniels with and without Chiari like Malformation-Syringomyelia Syndrome. They provided evidence for correlation of clinical symptoms of CM-SM and two DTI parameters, fractional anisotropy (FA) and apparent diffusion coefficient (ADC). There are several points need to be addressed and improved, listed as follows.

1.     Clinical relevance of ADC and FA to CM-SM has been reported in human patients previously, which significantly comprises novelty of the current manuscript carried out in dogs. 

2.     FA and ADC values were statistically significant only in asymptomatic vs symptomatic dogs already diagnosed with CM-SM while no difference in control vs diagnosed groups. How to reconcile this discrepancy in terms of utilizing FA and ADC as pre-diagnostic parameters?

Author Response

FA and ADC values were statistically significant only in asymptomatic vs symptomatic dogs already diagnosed with CM-SM while no difference in control vs diagnosed groups. How to reconcile this discrepancy in terms of utilizing FA and ADC as pre-diagnostic parameters?

Dear Reviever,

Thank You for Your assessment, comment and question

Our research is a preliminary study, in which we wanted to asses the usefulness oft he DTI-method for the assessment of CM-SM syndrome in CKCS due to the problem of lack of correlation between a clinical symptoms and radiological changes. We belive that that the method can be used to asses the risk of developing clinical symptoms. We’ve already found a statistically significant sifferences between dogs with symptoms and without them. We did not find a statistically significant correlation between the nonSM and SM groups, however, there was a tendency towards reduced values of FA and increased values of ADC in dogs with SM compared to the nonSM group. In our opinion, decreased FA values and increased ADC values with  correlates with the possibility of development syringomyelia.

In clinical practice, syringomyelia does not need to be treated for patients with no neurological symptoms or signs, even if the syringomyelia is very large. However, in some patients without clinical symptoms, the microstructural function of the spinal cord at the cavity may have been seriously disturbed, which requires timely intervention and treatment.

Of course it is necessary to conduct further research on a larger group of animals.

Reviewer 3 Report

Dear editor

·       Thanks for the invitation to review the manuscript entitled (Diffusion Tensor Imaging in Syringomyelia secondary to Chiari Malformation in Cavalier King Charles Spaniel – a preliminary study.). The paper is interesting which highlight the relationship between new imaging techniques in the disease for clinical setting regarding to different forms. However, substantial editing id needed before accepting the paper for publication in Animals.

·       The following are my comments to enhance the paper.

In abstract:

what do you mean by same protocol? do you mean the same protocol for all dogs or same as previous reported>

which neurological test used?

summarize some numerical data and add it in the last paragraph of the abstract

please add a conclusion or suggestion of your result and how it will be benefit in clinical studies.

Introduction:

Line 105: use in the clinical setting.

Line 106: it should be in the conclusion not in introduction.

M & M

Line 155: did you examine the interobserver variability? it is important to mention it.

Figure 1: please add topographic details on the images. Animals journal is not a neurologic journal. please add the details and define exactly the region of interest used in your scaling. define the surrounding structures. Also, did the ROI in all cases are in the same position. Must be identified.

Line 190 how did you define the ROI. add ref

Line 192: why 3 ROI only? add reference

Statistical analysis:

Significant editing must be performed in the statistical analysis section (writing the full details of data and design) and more methods should be added

Normality testing, interobserver variability and quality of score, CV (horizontally between 3 ROI and vertically between all cases, which type of T test and ANOVA for parametric or non-parametric, which post hoc test, correlation is not enough please add regression analysis

Results:

CKCS with MRI confirmed SM (SM group) (age 6–70 months, 36 weight 3.5–9.1 kg, M/F 6/12) and CKCS dogs without SM (non-SM group) (7–51 months, weight 37 4–9.5 kg, M/F 5/7). According to this classification your statistical design includes three groups, and you should show the result of comparison between three groups not only SM symptomatic and non-symptomatic (I mean in tables). This has been illustrated in figure 6 but not in tables 1,2.

one more table describing the interobserver variability, CV, is needed.

Line 120: write details on the clinical symptoms and neurological testing used in your investigation with reference

thank you 

Author Response

Dear Reviever,

thank Your for all Your comments and advices. We’ve tried to change our manuscript according to them. All changes in the text in our manuscript are in green colour.

what do you mean by same protocol? do you mean the same protocol for all dogs or same as previous reported

We mean the same protocol for all dogs.

which neurological test used?

We performed the neurological examination consisting of mental status and behavior assessment, attitude/posture, gait, abnormal movements, postural reactions (proprioceptive positioning, hopping, hemiwalking), cranial nerves assessment, spinal reflexes (withdrawal reflex, patellar reflex, cranial tibial reflex, extensor carpi radialis reflex, perineal reflex, cutaneous trunci reflex), palpation of a head, neck and spinal cord.

summarize some numerical data and add it in the last paragraph of the abstract

Dear Reviewer,

 You asked us to add some numerical values in our abstract and Reviewer 1 asked us to make our abstract more specific. We tried to find a compromiss and we hope it will work.

please add a conclusion or suggestion of your result and how it will be benefit in clinical studies.

We changed it in manuscript.

Introduction:

Line 105: use in the clinical setting.

To our knowledge this is the first report evaluating DTI parameters in dogs with syringomyelia use in the clinical setting – we changed it in manuscript.

Line 106: it should be in the conclusion not in introduction.

As above

M & M

Line 155: did you examine the interobserver variability? it is important to mention it.

Measurements were made by one person, usually two measurements were made, which did not differ from each other. We paid a lot of attention to ensure that the ROIs were drawn properly and covered only the area of spinal cord.

Before starting the research, we’ve learned how to use this method and we did not observe differences between the measurements made by AB, KOS.

Figure 1: please add topographic details on the images. Animals journal is not a neurologic journal. please add the details and define exactly the region of interest used in your scaling. define the surrounding structures. Also, did the ROI in all cases are in the same position. Must be identified.

Thank you for the comment. As it is stated in the manuscript the MRI protocol imaging as far as DTI were predefined, ROI in all cases was in this same position. For the figures we added the identification marks to the MRI images for the better position understanding.

ROI in all cases was in this same position.

Line 190 how did you define the ROI. add ref

As above

Line 192: why 3 ROI only? add reference

ADC and FA values were not measured within the thoracic spinal cord because of the greater susceptibility to artifacts in these regions (27). (line 209)

We suspected to have the biggest change in the region of CM that's why we examined just three ROI's. The other reason is a duration of MRI examination. We had to choose the most interesting ROI, otherwise the examination will be to long for animals. 

Statistical analysis:

Significant editing must be performed in the statistical analysis section (writing the full details of data and design) and more methods should be added

Following the reviewer's comments, the section on statistical analysis has been described in more detail.

Testing for the normal distribution of the data obtained was performed with the Shapiro–Wilk normality test. The assumptions about the normality of the data distribution were maintained in the analyzed cases (p>0.05).

The obtained FA and ADC values are parametric values, therefore statistical tests for parametric values were used (Student's t-test, ANOVA).

Before using ANOVA, Leven's test was performed to assess the homogeneity of variance in the analyzed cases. In all analyzed cases, the homogeneity of variance was maintained (p>0.05), therefore ANOVA could be used.

The word ‘correlation’ has been used to illustrate the relationship. Clinical symptoms are not parametric values, it is not possible to carry out a linear correlation of the obtained results with clinical symptoms. Therefore the description of statistical methods of correlation analysis with the analysis of the regression coefficient is not mentioned because we do not see the possibility of using these methods in the study.

We’ve changed a word correlation for tendency/dependency. It could be misleading.

Results:

CKCS with MRI confirmed SM (SM group) (age 6–70 months, 36 weight 3.5–9.1 kg, M/F 6/12) and CKCS dogs without SM (non-SM group) (7–51 months, weight 37 4–9.5 kg, M/F 5/7). According to this classification your statistical design includes three groups, and you should show the result of comparison between three groups not only SM symptomatic and non-symptomatic (I mean in tables). This has been illustrated in figure 6 but not in tables 1,2.

We changed it in manucript

one more table describing the interobserver variability

Measurements were made by one person, usually two measurements were made, which did not differ from each other. We paid a lot of attention to ensure that the ROIs were drawn properly and covered only the area of spinal cord.

Before starting the research, we’ve learned how to use this method and we did not observe differences between the measurements made by AB, KOS.

Line 120: write details on the clinical symptoms and neurological testing used in your investigation with reference

Clinical symptoms have been already described (lines 124-128 and 251-253)

Based on interviews with the owners and neurological examination, the dogs were as signed to the SM-symptomatic or SM-asymptomatic groups. The following symptoms were reported: abnormal behaviour, “phantom scratching”, fly-catching, ear flapping, 126 neuropathic pain, and neurological deficits such as thoracic limb weakness and muscle  atrophy, as well as pelvic limb ataxia and weakness.

Eight dogs out of 10 in the SM group were symptomatic (SM-symptomatic group), and showed neurological deficits (5/8 showed weakened flexor reflex on front limbs, delayed proprioception, neck pain and 3/8 showed weakened flexor reflex on front limbs, neck pain and delayed proprioception). (251-253)

Round 2

Reviewer 1 Report

Dear authors, thank you for addressing my comments and editing the text when necessary. Some final ones to check.

Line 14: the severity...does not commensurate (rather than is not)

Line 36: the same clinical...

Line58; fluid-filled

Line 48: you did not grade the severity of the clinical signs, therefore you cannot say that MR findings are correlated to their severity. You could only say that the DTI parameters correlate to the presence of clinical findings.

Line 138: it can suffice to say European boarded neurologist or similar.

Reviewer 3 Report

thank you for addressing our comments.

in abstract, the author mentioned that the data was correlated with clinical symptoms. again please correct in the final revision.